# Co-Doped NdFeO_3_ Nanoparticles: Synthesis, Optical, and Magnetic Properties Study

**DOI:** 10.3390/nano11040937

**Published:** 2021-04-06

**Authors:** Tien Anh Nguyen, Thanh Le Pham, Irina Yakovlevna Mittova, Valentina Olegovna Mittova, Truc Linh Thi Nguyen, Hung Van Nguyen, Vuong Xuan Bui

**Affiliations:** 1Informetrics Research Group, Ton Duc Thang University, Ho Chi Minh City 700000, Vietnam; nguyenanhtien@tdtu.edu.vn; 2Faculty of Applied Sciences, Ton Duc Thang University, Ho Chi Minh City 700000, Vietnam; 3Faculty of Chemistry, Ho Chi Minh City University of Education, Ho Chi Minh City 700000, Vietnam; thanhhoahoc@gmail.com (T.L.P.); linhntt@hcmue.edu.vn (T.L.T.N.); 4Faculty of Chemistry, Voronezh State University, Voronezh 394018, Russia; imittova@mail.ru; 5Department of Biochemistry, Voronezh State Medical University named after N. N. Burdenko, Voronezh 394036, Russia; vmittova@mail.ru; 6Practice and Experimental Center for Dong Thap University, Cao Lanh City 81000, Vietnam; nguyenvanhung@dthu.edu.vn; 7Faculty of Pedagogy in Natural Sciences, Sai Gon University, Ho Chi Minh City 700000, Vietnam

**Keywords:** nanocrystals, co-doped NdFeO_3_, co-precipitation method, optical properties, magnetic properties

## Abstract

In this work, single-phase nanostructured NdFe_1−*x*_Co*_x_*O_3_ (*x* = 0, 0.1, 0.2, and 0.3) perovskite materials were obtained by annealing stoichiochemistry mixtures of their component hydroxides at 750 °C for 60 min. The partial substitution of Fe by Co in the NdFeO_3_ crystal lattice leads to significant changes in the structural characteristics, and as a consequence, also alters both the magnetic and optical properties of the resulting perovskites. The low optical band gap (*E*_g_ = 2.06 ÷ 1.46 eV) and high coercivity (*H*_c_ = 136.76 ÷ 416.06 Oe) give Co-doped NdFeO_3_ nanoparticles a huge advantage for application in both photocatalysis and hard magnetic devices.

## 1. Introduction

Perovskite orthoferrite materials have been intensively studied because of their diverse unique properties, variable formulae, variable structures, and wide technological applications. Especially, the substitution of Ln or Fe sites of the LnFeO_3_ perovskite-type oxides (where Ln is a rare-earth element such as La, Nd, and Pr) by other elements which can exhibit considerable multi-valence and defect sites in their structures, which lead to the tunable redox and electromagnetic characteristics of the materials [1,2,3]. In order to boost the performances and widen the applications of perovskite materials, much effort has been spent to downscale the structure into nano size. Indeed, in comparison with their bulk counterparts, perovskite nanomaterials show many advantages, such as high processing capability of thin film [4], rich and controllable catalyst active sites [5,6] or excellent optical, electrical, and magnetic properties [7,8,9].

In NdFeO_3_, the magnetic moments of Fe and Nd are two antiparallel coupled nonequivalent magnetic sublattices. The electrons in the 3d and 4f orbitals of these two sublattices interact with spin-lattice coupling, leading to a very unstable magnetic state and, thus, they result in unusually large magnetic anisotropy, magnetization reversal, and spin switching in low magnetic fields [10]. In order to adjust their optical, electrical, and magnetic properties and to enhance the performance of pristine perovskite, transition metal ions can be doped into their crystal lattices. Co-doped LnFeO_3_ materials have attracted extensive study thanks to their exciting dielectric, sensing, optical, and magnetic properties. Owing to the multiple spins and oxidation states of Co, the process of catalysts may be modified by changing the concentration of Co in solid solutions [3,6,11,12,13,14].

Several studies [15,16,17,18,19,20] described the formation of LnFeO_3_ orthoferrites nanoparticles (Ln = La, Y), including those doped with metals (for example, Mn, Co, Ni, and Ba) by a simple co-precipitation method via the hydrolysis of cations in boiling water followed by the addition of appropriate precipitants. In our recent work [21], NdFeO_3_ nanoparticles, of 30 nm in size, were obtained via the simple co-precipitation method mentioned above, their crystal structure and magnetic properties were also studied therein.

In this paper, single-phase Co-doped NdFeO_3_ nanoparticles were synthesized and the changes in their crystal structure, their magnetic and optical properties were also studied. To the best of our knowledge, similar work has not been reported elsewhere.

## 2. Materials and Methods

All reagents in this work are analytical grade and were used without any further purification. The procedure for synthesizing Co-doped NdFeO_3_ nanoparticles is similar to that of NdFeO_3_ [21], with NaOH 5% as the precipitant instead of a NH_3_ 5% solution, in order to avoid the generation of soluble complex from the reaction of cobalt (II) hydroxide precipitate (Co(OH)_2_↓) and ammonium solution according to Equation (1) [22].
Co(OH)_2_↓ + 6NH_3_ → [Co(NH_3_)_6_](OH)_2_(1)

The structure and phase composition of the samples were investigated by X-ray powder diffraction (XRD, D8-ADVANCE, Brucker, Bremen, Germany) with Cu K_α_ radiation (*λ* = 1.54056 Å), the step size is chosen to be 0.02 in range of 10° to 80°. The average crystal size was determined according to the Debye–Scherrer equation [23]; lattice constants *a*, *b*, *c*, and the unit cell volume *V* were determined using the Rietveld method [23] implemented in the X’pert High Score Plus 2.2b software package [18,19,20].

The energy-dispersive X-ray spectroscopy (EDX) was carried out with a FE-SEM S-4800 spectrometer (Hitachi, Tokyo, Japan). The average value of five different positions in each sample was taken as the final result of the corresponding sample. Particle size and morphology of Co-doped NdFeO_3_ nanoparticles were determined using transmission electron microscopy (TEM; JEOL-1400, Jeol Ltd, Tokyo, Japan).

The UV-Vis absorption spectra of NdFe_1−*x*_Co*_x_*O_3_ nanocrystals were studied on a UV-Visible spectrophotometer (UV-Vis, JASCO V-550, Shimadzu, Tokyo, Japan). The optical energy gap (*E*_g_, eV) was determined by fitting the absorption data to the direct transition as in previous publication [24].

Magnetic properties of the samples (the saturation magnetization *M*_s_ in the maximal field, the coercive force *H*_c_ and remanent magnetization *M*_r_) were investigated at 300 K via a vibrating sample magnetometer (VSM, MICROSENE EV11, Tokyo, Japan).

## 3. Result and Discussion

### 3.1. Structures and Morphologies of Nanostructured NdFe_*1*−x_Co_x_O_3_

The XRD patterns of the NdFe_1−*x*_Co*_x_*O_3_ products (*x* = 0, 0.1, 0.2, 0.3, and 0.4) compared with those of the Nd_2_O_3_, Fe_2_O_3_, and Co_3_O_4_ component oxides independently prepared under similar conditions (annealed at 750 °C during 60 min) are shown in Figure 1. The annealing condition was determined according to the previous work [21]. Those patterns confirmed that the NdFe_1−*x*_Co*_x_*O_3_ samples with *x* = 0, 0.1, 0.2, and 0.3 were pure orthoferrite phase perovskite structure (NdFeO_3_, JCPDS No. 01-074-1473), with no identified peaks of oxide impurities. Interestingly, Co_3_O_4_ oxide was obtained instead of CoO since Co(OH)_2_ hydroxide can be oxidized and decomposed after annealing at high temperature [24].

In the case of *x* = 0.4, aside from the peaks corresponding to the NdFeO_3_ phase, there were peaks of the Nd_2_O_3_ phase (JCPDS No. 00-041-1089) at 2*θ* = 25.71 and 30.74°, and Co_3_O_4_ phase (JCPDS No. 00-043-1003) at 2*θ* = 38.87°. Thus, the successful substitution of Co into NdFeO_3_ crystal structures only took place when *x* was less than 0.4. With the increase in Co concentration, the XRD peak shifted toward a higher 2*θ* (right shift) and gradually broadened while the intensity of peaks decreased. Consequently, there was a decrease in unit cell volume (from V = 238.56 to V = 233.29 Å^3^) and in crystal size (from D_XRD_ = 28 ± 5 to D_XRD_ = 19 ± 3 nm) Figure 2 and Table 1. Such a decrease is also a confirmation for the Co (III) substitution to Fe (III) in the NdFeO_3_ crystal lattice. The substitution of Fe^3+^ ions (*r*_Fe3+_ = 0.65 Å [24]) by smaller Co^3+^ ions (*r*_Co3+_ = 0.55 Å [24]) led to the reduction of the unit cell parameters and crystal size following Vegard’s law, in which lattice parameters linearly varies with the degree of substitution of atoms or ions by others in ideal solid solution. The similar results were published in the previous research [12,13,18].

TEM images and particle size distribution for NdFe_1−*x*_Co*_x_*O_3_ samples (*x* = 0.1, 0.2, 0.3) are shown in Figure 3. As can be seen, the shape of the particles of the synthesized NdFe_1−*x*_Co*_x_*O_3_ samples is close to spherical, but agglomerates of particles are noticeable. For the NdFe_0.9_Co_0.1_O_3_ sample with the lowest level of cobalt doping, the particle size was in the range of 10–80 nm. The average particle diameter was 47 ± 5. For the other two samples (NdFe_0.8_Co_0.2_O_3_ and NdFe_0.7_Co_0.3_O_3_), the size of most particles was in the range of 20–70 nm. An analysis of the results of the size distribution of NdFe_1−*x*_Co*_x_*O_3_ particles Figure 3 allows us to conclude that the average crystallite size decreases monotonically with an increase in the dopant content in the synthesized samples. The lower values of D_avg_ calculated based on the XRD data as compared to the TEM results were due to the peculiarities of the used methods. The determination of the average crystallite size by the calculation method according to the Debye–Scherrer formula leads to significant errors that can be caused by the choice of a mathematical model for analyzing the X-ray line profile for the determination of the particle size and the influence of various factors on the broadening effect of diffraction maxima. In addition, the diffraction method is volumetric and therefore determines the size of crystallites averaged over the entire volume, in contrast to electron microscopy, which is a local visual method for estimating the size of particles (not crystallites) [25]. TEM results, to a certain extent, depend on the possibility of investigating only a relatively small number of particles under real conditions and on the quality of preliminary dispersion of nanopowders, which introduces a certain amount of uncertainty into the obtained results. Nevertheless, transmission electron microscopy is a direct and accurate method for determining the size and shape of nanoobject particles.

### 3.2. Elemental Composition of NdFe_*1−*x_Co_x_O_3_ Samples

According to the EDX results, the composition of the obtained NdFe_1−*x*_Co*_x_*O_3_ samples included only Nd, Fe, Co, and O, and as the concentration of cobalt ions in the initial solutions increased, their content in the NdFe_1−*x*_Co*_x_*O_3_ samples increased Table 2. From Table 2 it follows that the real content of each element in the synthesized samples is quite close to their nominal composition.

### 3.3. Optical and Magnetic Properties of Nano-Structured NdFe_*1−*x_Co_x_O_3_ (x = 0, 0.1, 0.2, and 0.3) Materials

The magnetic and optical characterizations of the NdFe_1−*x*_Co*_x_*O_3_ (*x* = 0, 0.1, 0.2, and 0.3) nanomaterials (annealed at 750 °C for 60 min) were carried out at room temperature. The results prove that beside the structure, the substitution of Co in the NdFeO_3_ crystal lattice also impressively change the magnetic and optical properties of the samples Table 3 and Figure 4 and Figure 5.

Indeed, when the concentration of Co ions in NdFeO_3_ crystal lattice increased, all magnetic parameters, including *H*_c_ (258.22–416.04 Oe), *M*_s_ (0.93–0.98 emu/g), and *M*_r_ (0.13–0.18 emu/g) increased with the rise of Co concentration in the NdFeO_3_ lattice. In addition, these values were significantly higher than those of the original NdFeO_3_ material [21] (with the exception of *M*_r_). It can be explained by the fact that the substitution of Co ions into the NdFeO_3_ lattice can intensify the magneto-crystalline anisotropy. Besides, Co substitution also led to a change in Fe–O–Fe angles, as well as the oxidation of a small amount of Fe^3+^ ions to Fe^4+^ ions, to compensate for the charge caused by the appearance of Co^2+^ at the sites of Fe^3+^. The similar phenomenon was also reported for HoFe_1−*x*_Ni*_x_*O_3_ [26], NdFe_1−*x*_Ni*_x_*O_3_ [27], GdFe_1−*x*_Ni*_x_*O_3_ [28], YFe_1−*x*_Co*_x_*O_3_ [18], and LaFe_1−*x*_Ni*_x_*O_3_ series [19]. Remarkably, under the same synthesis conditions, the magnetic parameters, especially *H*_c_, of the NdFe_1−*x*_Co*_x_*O_3_ nano-crystalline perovskite oxides are higher than those of other perovskite oxides, such as YFe_1−*x*_Mn*_x_*O_3_, YFe_1−*x*_Co*_x_*O_3_, and LaFe_1−*x*_Ni*_x_*O_3_ [17,18,19]. From those results, the magnetic properties of perovskite-type nanostructured materials can be easily tuned by varying the element and the degree of doping. This important feature of nanosized perovskites give them a wide variety of application in many different fields of magnetic materials.

The UV-Vis absorption spectra of the Co-doped NdFeO_3_ nanoparticles showed strong absorption in the ultraviolet (~300–400 nm) and visible light regions (~400–600 nm) Figure 4a. As the concentration of Co ions increased, there was a red-shift in the UV-Vis absorption spectra (toward the visible light region). The optical energy gaps (*E*_g_, eV) of the NdFe_1−*x*_Co*_x_*O_3_ nanomaterials (*x* = 0, 0.1, 0.2, and 0.3) were calculated similarly to other publications [24,26] and are shown in Table 2 and Figure 4b. The estimated direct band gaps of all products are in the range of 2.06–1.46 eV and increase when Co content in NdFeO_3_ lattice increases. Particularly, the Co-doped NdFeO_3_ nanoparticles in this work exhibited much narrower band-gap compared to some other related orthoferrites synthesized by other methods. For instance, the direct band-gaps of NdFe_1−*x*_Co*_x_*O_3_ (*x* = 0–0.4) and HoFe_1−*x*_Ni*_x_*O_3_ (*x* = 0–0.5) nanoparticles were reported to be 3.35 ÷ 3.04 and 3.39 ÷ 3.28 eV, respectively [26,29], and the values for LaFe_1−*x*_Ti*_x_*O_3_ (*x* = 0.2 ÷ 0.8) nanoparticles prepared by co-precipitation technique were 2.05–2.61 eV [2]. The small band gaps of NdFe_1−*x*_Co*_x_*O_3_ can give an advantage for the application of this material series in photocatalysis, gas sensor, and electrode materials in solid oxide fuel cells [12,13,29,30,31].

## 4. Conclusions

The single-phase nanostructured NdFe_1−*x*_Co*_x_*O_3_ (*x* = 0, 0.1, 0.2, and 0.3) perovskites have been synthesized by the simple co-precipitation method. The maximum level of substitution of iron with cobalt, which was *x* < 0.4 (XRD) was established. At *x* = 0.4, the homogeneity region was impaired and a phase mixture, consisting of Nd_2_O_3_ and Co_3_O_4_ was formed.

Obtained Co-doped NdFeO_3_ nanoparticles, after annealing at 750 °C for 60 min, have their crystal size (*D*_XRD_ = 25 ± 3 ÷ 19 ± 3 nm, D_TEM_ = 47 ± 5 ÷ 42 ± 3 nm), unit cell volume (*V* = 238.56 ÷ 233.29 Å^3^).

The study of the effect of the degree of substitution in NdFe_1−*x*_Co*_x_*O_3_ crystals on their optical and magnetic characteristics showed that optical band-gap values (*E*_g_ = 2.06 ÷ 1.46 eV) decreased while the coercive force (*H*_c_ = 136.76 ÷ 416.06 Oe) and saturation magnetization (*M*_s_ = 0.80 ÷ 0.98 emu/g) increased with the increase of Co content. Co-doped NdFeO_3_ nanoparticles have low optical energy gaps and high coercivity, which are beneficial not only for application in photocatalysis, but also for hard magnetic devices (permanent magnets or recorders).

## Figures and Tables

**Figure 1 nanomaterials-11-00937-f001:**
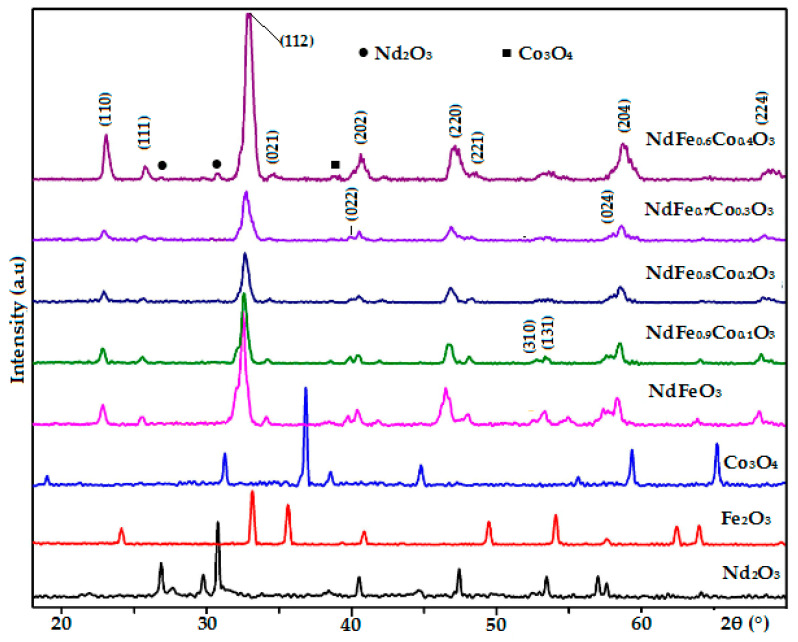
X-ray powder diffraction (XRD) patterns of NdFe_1−*x*_Co*_x_*O_3_ samples (*x* = 0, 0.1, 0.2, 0.3, and 0.4) and Nd_2_O_3_, Fe_2_O_3_, and Co_3_O_4_ annealed at 750 °C for 60 min.

**Figure 2 nanomaterials-11-00937-f002:**
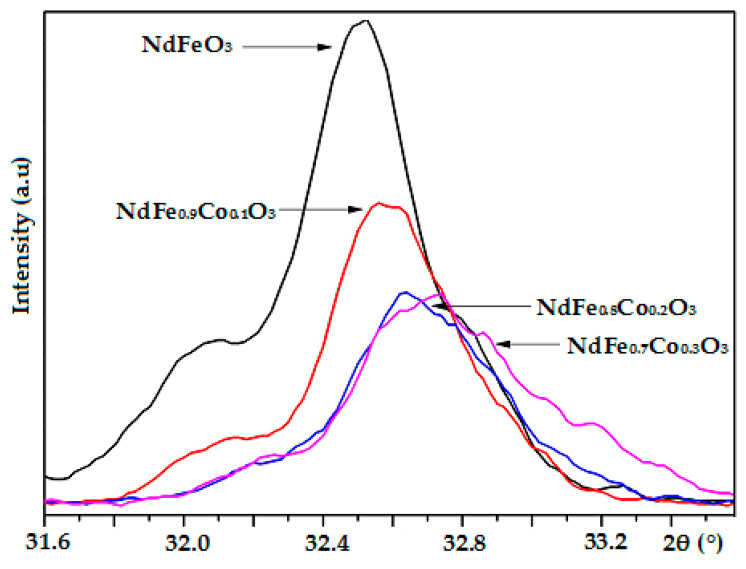
Slow-scan XRD patterns of peak (112) of NdFe_1−*x*_Co*_x_*O_3_ samples annealed at 750 °C for 60 min.

**Figure 3 nanomaterials-11-00937-f003:**
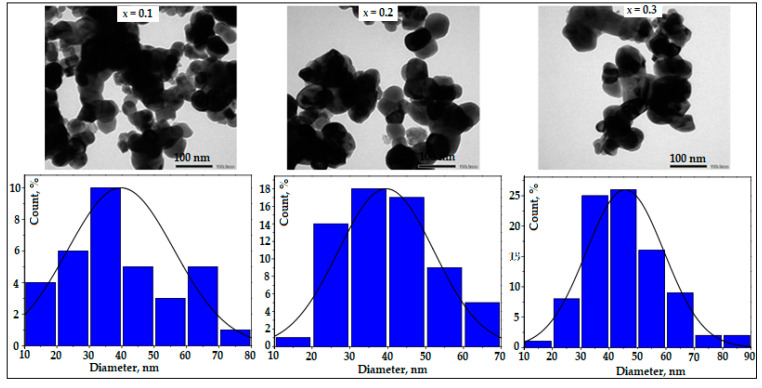
Transmission electron microscopy (TEM) images and particle size distribution of NdFe_1−*x*_Co*_x_*O_3_ samples annealed at 750 °C.

**Figure 4 nanomaterials-11-00937-f004:**
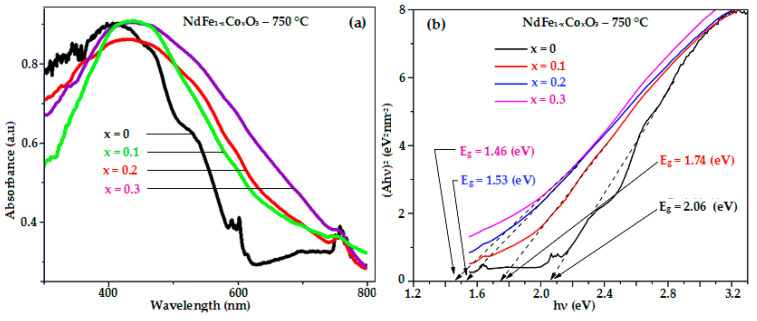
(**a**) Room-temperature optical absorbance spectra and (**b**) plot of (Ahν)^2^ as a function of photon energy for NdFe_1−*x*_Co*_x_*O_3_ materials annealed at 750 °C for 60 min.

**Figure 5 nanomaterials-11-00937-f005:**
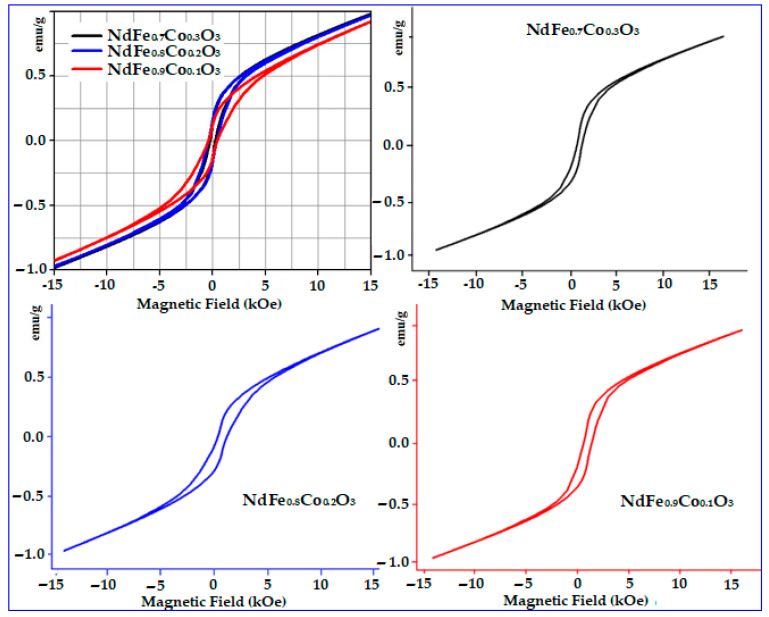
Field dependence of the magnetization of NdFe_1−*x*_Co*_x_*O_3_ nanoparticles annealed at 750 °C for 60 min.

**Table 1 nanomaterials-11-00937-t001:** Lattice parameters and crystallite sizes of NdFe_1−*x*_Co*_x_*O_3_ samples annealed at 750 °C for 60 min.

NdFe_1−*x*_Co*_x_*O_3_	2θ^o^ (112)	D_avg_, nm		Lattice Constants, Å			*V*, Å^3^
		XRD	TEM	*a*	*b*	*c*	
NdFeO_3_ [21]	32.49	28 ± 5	-	5.4990	5.5910	7.7592	238.56
NdFe_0.9_Co_0.1_O_3_	32.54	25 ± 3	47 ± 5	5.4257	5.5919	7.7638	235.55
NdFe_0.8_Co_0.2_O_3_	32.57	22 ± 2	45 ± 6	5.4425	5.5292	7.7616	233.57
NdFe_0.7_Co_0.3_O_3_	32.61	19 ± 3	42 ± 3	5.4113	5.5504	7.7673	233.29

**Table 2 nanomaterials-11-00937-t002:** EDX results of NdFe_1−*x*_Co*_x_*O_3_ samples annealed at 750 °C for 60 min.

Nominal Composition of Samples	Elemental Composition (at. %)				Real Composition of Samples
	Nd	Fe	Co	O	
NdFeO_3_	18.25 ± 1.57	20.45 ± 1.11	0.00	61.30 ± 2.17	NdFe_1.120_O_3.359_
NdFe_0.9_Co_0.1_O_3_	18.95 ± 1.03	18.45 ± 1.07	1.43 ± 0.17	61.17 ± 2.36	NdFe_0.973_Co_0.075_O_3.228_
NdFe_0.8_Co_0.2_O_3_	19.01 ± 1.42	16.03 ± 0.89	2.13 ± 0.35	62.83 ± 3.21	NdFe_0.843_Co_0.112_O_3.305_
NdFe_0.7_Co_0.3_O_3_	19.27 ± 1.35	14.31 ± 0.73	5.41 ± 0.42	61.01 ± 3.08	NdFe_0.743_Co_0.281_O_3.166_

**Table 3 nanomaterials-11-00937-t003:** Optical and Magnetic characteristics of NdFe_1−*x*_Co*_x_*O_3_ nanomaterials annealed at 750 °C for 60 min.

NdFe_1−*x*_Co*_x_*O_3_	*H*_c_, Oe	*M*_r_, emu/g	*M*_s_, emu/g	*E*_g_, eV
NdFeO_3_ [21]	136.76	0.68	0.80	2.06
NdFe_0.9_Co_0.1_O_3_	258.22	0.13	0.93	1.74
NdFe_0.8_Co_0.2_O_3_	395.79	0.15	0.97	1.53
NdFe_0.7_Co_0.3_O_3_	416.06	0.18	0.98	1.46

## Data Availability

Not Applicable.

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
