# Peer review of "Co-Doped NdFeO3 Nanoparticles: Synthesis, Optical, and Magnetic Properties Study"

_nanomaterials, 2021, doi:10.3390/nano11040937_

Round 1

Reviewer 1 Report

The authors report on the ease preparation of Co-doped NdFeO3 nanoparticles. The doping resulted in changes in the structural features and, as a consequence, in their optical and magnetic properties. TEM analysis demonstrate the formation of large clusters and practically no isolated nanoparticles are detected; this is a drawback that the authors should solve. The photoluminescence quantum yield of the samples are relevant data to be also included in the paper.  

In my opinion, the paper is not ready to be considered for publication in Nanomaterials.

Author Response

Dear Professor,

We tried very hard to correct the article. We would like to thank your comments.

Best Regards,

Reviewer 2 Report

The NdFeO3 is an interesting compound, the authors put in evidence possibility of tuning the magnetic and optical properties by Co doping. The results can be interesting, but the work seems wrote in hurry and need a strong revision.

REVIEW

  • Explain the difference in grain size between XRD and TEM measurements. In line 95 the authors claim that the average size by TEM is 30-50 nm. Is this measurement referred to all the samples? If the authors mean that the average value spans from 30 to 50 nm, in this case the measure seems very inaccurate. Generally, the difference in size valuation depends on the crystallized part of grains that is detected by XRD. It is lower that the real particle dimension that includes the possible amorphous border. In my opinion High resolution TEM investigation would be interesting to check the crystallization at the particle border. At the same time, it would be interesting to perform selected area electron diffraction.
  • Page 4 - change Table 1 into table 2
  • Figure 3 is not clear, please put better in evidence the markers
  • The observed particle aggregation can increase the magnetic interaction and affect the Hc, Ms and Mr, what is the authors opinion about this? Moreover, would be interesting to compare the hysteresis loops with the not doped one.
  • Figure 6 - In my opinion it is better to superimpose the hysteresis loops for a better comparison
  • Cut figure 4, as it does not contain useful information itself, on the other hand it is mandatory to put in a table the results of the elements quantified by EDX
  • How did the authors estimated Ms ?
  • Some typos in the text…
  • Line 123- change figure 1 into figure 6
  • Put the measurement errors.
  • Improve the conclusions: they are very poor and not clear

Author Response

(The authors gave the same response as above.)

Round 2

Reviewer 1 Report

The authors have addressed the suggested revisions satisfactorily. 

Reviewer 2 Report

The authors replied satisfactorily to the comments